# PREFERENCE-BASED CREDIT ASSIGNMENT FOR REINFORCEMENT LEARNING WITH DELAYED AND NOISED REWARDS

## ABSTRACT

Credit assignment has been utilized as a common technique for determining the past key state-action pairs and assigning corresponding rewards that have strong relevance with the final outputs in reinforcement learning, especially for environments with delayed rewards. However, current reward function design methods rely heavily on domain knowledge and may not accurately reflect the actual reward that should be received, which will lead to noised reward assignments during the credit assignment process and deteriorate the performance of the agent. To address this issue, in this paper, by leveraging the benefits of Preference-based Reinforcement Learning (PbRL), we propose a novel trajectory preference-based credit assignment method, where each trajectory is assigned to one of three different preferences according to its related delayed reward and the entire trajectory space. Then, a causal Transformer framework is introduced to predict the relevance between the decisions at each timestep and the different trajectory preferences to guide the credit assignment. Despite the unavoidable noised reward related to each trajectory, we demonstrate that our method can still effectively guide agents to learn superior strategies. Experiments on the MuJoCo task and the treatment of sepsis under extremely delayed reward setting show that our method can mitigate the adverse effects resulting from the delayed noised rewards and provide effective guidelines for agents.

## 1 INTRODUCTION

Reinforcement learning (RL) has achieved impressive performances in many domains, such as video games (Mnih et al., 2013), the game of Go (Silver et al., 2017), InstructGPT (Ouyang et al., 2022), among others. While these remarkable applications demonstrate the significant potential and research value of RL, its performances rely heavily on well-designed reward functions (Gangwani et al., 2020). However, in real-world sequential decision-making problems, rewards are usually sparse and/or delayed (Ren et al., 2022). For example, in the heparin dosing therapy for Intensive Care Unit (ICU) patients, activated partial thromboplastin time (aPTT), which is a key evaluation criterion and reward signal of the effects of the treatment, can only be acquired after 4 to 6 hours of the intravenous administration (Liang et al., 2023). In extreme cases, i.e., episodic reinforcement learning (Han et al., 2022), the rewards are only given at the final state and are the sole evaluation metric for past decisions. This makes it difficult for RL agents to determine the relevance between final feedback and past decisions, leading to low learning efficiency (Chen & Tan, 2023).

To address this episodic reward issue, heuristic reward functions are designed for generating dense rewards (Andrychowicz et al., 2017). However, even well-designed rewards may be inaccurate and can mislead RL agents into undesirable behaviors. As an alternative, exploration and credit assignment related techniques have been proposed and are seen as more effective solutions, where exploration can help agents to find better behaviors (Ménard et al., 2021), and credit assignment can assign rewards to the past relevant decisions (Zhang et al., 2024).

With the applications of reinforcement learning in more real-world problems, where the long-term credit assignment problem and the sparse reward problem co-exist, it becomes harder for exploration techniques to be successfully used as in such domains, current exploration methods may fail to

discover the key actions that have the most influence on the final output due to the delayed reward signals (Chen et al., 2023), leading to decreased performances and lower learning efficiency of the RL agent. In theory, credit assignment approaches can mitigate this problem in a more favorable way, and based on return decomposition, existing credit assignment methods (Zhang et al., 2024; Ren et al., 2022; Gangwani et al., 2020) decompose episodic rewards by predicting rewards in the trajectory and then assigning them to past decisions. However, as mentioned, the designed episodic rewards are usually noisy and may not accurately reflect the actual reward that should be received. With the noisy episodic reward, the decomposition of it to past decision steps will also be inaccurate, and then the performances of the underlying agent will be affected.

In fact, rather than defining a reward function, it is easier to judge a trajectory as "good" or "bad" and provide qualitative labels to it. By utilizing this idea, Preference-based Reinforcement Learning (PbRL) (Novoseller et al., 2020) can learn from paired trajectory samples that contain such qualitative labels, in which trajectories are manually labeled as preferences of "good" or "bad" in pairs, reflecting the human's relative evaluation of one trajectory compared to the other one. However, current PbRL methods can only learn from manually labeled paired feedbacks, which makes these methods limited to specific offline datasets.

In this paper, by leveraging the benefits of PbRL, we propose a novel trajectory preference-based credit assignment method. Specifically, we first extend the concept of preferences by defining three preferences ("good", "neutral" or "bad") rather than paired preferences according to the trajectory's episodic reward and the entire trajectory space, and one of the three preferences will be assigned to each trajectory as a label to describe its quality. Then, a causal transformer framework is introduced to model trajectory sequences. By utilizing the transformer's self-attention mechanism, for each trajectory, the relevance between the decisions at each time step and its preference will be predicted and used as a guideline for assigning the trajectory's episodic reward to past decision steps. Overall, the main contributions are as follows:

1. A global preference setting mechanism is proposed according to episodic rewards and the entire trajectory space, which is used to describe the quality of each trajectory as "good", "neutral" or "bad", and could mitigate the adverse effects of noised rewards.

2. By introducing a causal transformer framework to model the temporal relationships among trajectory sequences, for each trajectory, the relevance between each decision step and its preferences could be predicted to guide credit assignment for agents. What's more, this framework also allows our method to accept trajectories of varying lengths as inputs.

3. With the aforementioned relevance as the guideline, even noisy episodic rewards can be assigned back to past decisions and the performance of the agent can still be guaranteed. Experimental results on the sepsis treatment offline dataset and the MuJoCo online environment PointMaze demonstrate the effectiveness of our method, which also show that compared to traditional credit assignment methods, our method offers a more general form of trajectory feedback that is robust to the noise of the episodic rewards.

## 2 RELATED WORK

**Credit assignment**   Credit assignment is one of the central challenges in reinforcement learning. Arjona-Medina et al. introduced a return decomposition formula for credit assignment where episodic rewards are decomposed into rewards for each time step by using an LSTM-based long-term return prediction model and a manually designed assignment rule (Holzleitner et al., 2021). In (Gangwani et al., 2020), based on return decomposition, the concept of uniform reward redistribution, namely IRCR, is proposed. In subsequent work, EMR (Chen et al., 2023) combines the intrinsic rewards of the exploration mechanism with IRCR to balance exploration and exploitation and avoid the predictive model falling into local optima. In (Ren et al., 2022), an upper bound for the common return-equivalent assumption is introduced to serve as a surrogate optimization objective, and return decomposition can be combined with the uniform reward redistribution framework in IRCR.While these methods have achieved good performances on many tasks, the defining of the episodic rewards used often requires rich prior knowledge of the underlying systems, and the set rewards may also contain noise, which will be cumulated during the credit assignment process and deteriorate the final performances of the agent.

**Preference-based reinforcement learning** In recent years, various methods have employed human preferences to train RL agents without reward engineering (Ouyang et al., 2022; Christiano et al., 2017; Ibarz et al., 2018; Kim et al., 2023). Christiano et al. (Ouyang et al., 2022) demonstrate that preference-based RL can effectively solve complex control tasks using deep neural networks, and the querying of human paired preferences is more efficient than querying from the demonstration data. Another advantage of PbRL is that humans can provide preferences about uncertainties to promote exploration efficiency (Liang et al., 2022). Despite these benefits, PbRL is ineffective in complex environments, as paired preferences only provide relative information rather than a direct evaluation of sample quality, and even if one sample is more preferred than the other, it does not necessarily mean that this sample is better (White et al., 2024). For PbRL, it is also challenging for humans to compare similar samples, which is time-consuming and may result in inaccurate preference labels. Unlike the relative comparison approach typically used in PbRL, in our approach, trajectories are evaluated from a global perspective according to the entire trajectory space and episodic rewards, which allows a qualitative classification of trajectories and reduces the adverse effects of uncertainty in preferences. Moreover, existing methods (Liu et al., 2022; White et al., 2024; Liang et al., 2022) mainly focus on using preference information to evaluate the overall quality of trajectories, and the problem of credit assignment within trajectories is not considered, which is validated to be effective in our paper.

**Transformer in reinforcement learning** Chen et al. (Chen et al., 2021) introduced transformer into offline RL, mainly taking advantage of the causal transformer sequence modeling to output decision actions end-to-end. Although this approach avoids the disadvantage of the traditional RL method of slow iteration through the Bellman equation in sparse reward scenarios, the framework is limited by the offline setting. Similar to our work is the Preference Transformer proposed by Kim et al. (Kim et al., 2023). This work uses the Causal Transformer to model human preferences and shows that the Causal Transformer can capture key events in trajectories and has the ability to assign long-term credits, but this work is limited to a specific human preference dataset. Compared with setting a reward function, labeling every two trajectory data undoubtedly increases the manual cost, while our work emphasizes learning from the manually set reward function by alleviating the impact of reward noise. Although the work of Ni et al. (Ni et al., 2024) showed that the Transformer lacks the ability to assign long-term credit, they only verified it on the ordinary Transformer, but not on the Causal Transformer with restricted attention order.

## 3 PRILIMINARIES

**Reinforcement learning** The environment is modeled as a Markov Decision Process (MDP) in reinforcement learning (Bellman, 1966), in which, at each time step $t$, the agent selects an action $a_t$ based on its current state $s_t$ and the policy $\pi$, and then it will receive a reward $r(s_t, a_t)$ and transition to the next state $s_{t+1}$. The process can be defined as $(S, A, \mathbb{P}, R, \rho, \gamma)$, where $S$ is the set of states, $A$ the set of possible actions, $\mathbb{P} : S \times A \times S \to [0, 1]$ the state transition probability function that represents the probability $p(s' \mid s, a)$ of reaching state $s' \in S$ after taking action $a$ in state $s$, $R$ the reward function, $\rho : S \to [0, 1]$ specifies the initial state distribution, and $\gamma \in [0, 1)$ is the discount factor, respectively. A trajectory $\tau = \{(s_t, a_t)\}_{t=1}^{T}$ will be generated by repeatedly executing a policy under an MDP, where $T$ is the length of the trajectory, and each trajectory $\tau$ is associated with the reward function $R(\tau)$ that we aim to optimize. The optimization objective is to learn to maximize $J(\theta) = \mathbb{E}_{\pi_\theta}[R(\tau)]$, where $R(\tau) = \sum_{t=1}^{\infty} \gamma^{t-1} r(s_t, a_t)$ is the discounted cumulative sum of rewards, and $\pi_\theta(a|s)$ is the conditional probability distribution of action $a$ given state $s$, with parameters $\theta \in \Theta$.

**Episodic reinforcement learning** Unlike the standard reinforcement learning framework, in episodic reinforcement learning, agents receive reward feedback only at the end of the trajectory (Chen et al., 2023), which measures the quality of the whole trajectory, i.e., the reward at each timestep $t$ is given as follows:

$$r(s_t, a_t) = \begin{cases} r_t, & \text{if } t < T \\ r_T, & \text{if } t = T \end{cases} \quad (1)$$

Then, the episodic reward for a trajectory of length $T$ is defined as: $R_{ep}(\tau) = \sum_{t=1}^{T} r(s_t, a_t) = r(s_T, a_T)$. Accordingly, the optimization objective in episodic reinforcement learning is transformed

to:

$$J_{ep}(\theta) = \mathbb{E}_{\pi_\theta}[R_{ep}(\tau)] = \mathbb{E}_{\pi_\theta}[r(s_T, a_T)] \tag{2}$$

It has been demonstrated that the environmental reward function in episodic reinforcement learning is non-Markovian, and the sample efficiency of reinforcement learning algorithms is significantly deteriorated.

**Preference-based reinforcement learning** In many applications, it is challenging to design a suitable and precise reward function. PbRL (Christiano et al., 2017) addresses this issue by learning reward functions from human preferences, where two trajectories of length $T$ ($\tau_0, \tau_1$) and $\tau = \{(s_1, a_1), \ldots, (s_T, a_T)\}$ are annotated by humans with preference labels. The preference labels are defined as $y \in \{0, 1, 0.5\}$, where $y = 1$ indicates $\tau_1 \succ \tau_0$, $y = 0$ indicates $\tau_0 \succ \tau_1$, $y = 0.5$ indicates equal preference, and $\tau_i \succ \tau_j$ means humans prefer trajectory $i$ to trajectory $j$. In the literature, most work (Christiano et al., 2017; Ibarz et al., 2018; Liang et al., 2022) estimates trajectory preferences using the Bradley-Terry model (Bradley & Terry, 1952), where the basic idea of this model can be roughly described by Equation (3) as follows:

$$P(A \succ B) = \frac{\exp(\beta_A)}{\exp(\beta_A) + \exp(\beta_B)} \tag{3}$$

where $P$ is the probability that trajectory $A$ beats trajectory $B$, and $\beta_A$ and $\beta_B$ are parameters associated with trajectories $A$ and $B$, reflecting their "strength" or preference level. For PbRL, to obtain the reward function parameterized by $\phi$, inspired by the Bradley-Terry model, existing work (Liang et al., 2022; Kim et al., 2023)

$$P(\tau_0 \succ \tau_1; \phi) = \frac{\exp(r(s_{t_0}, a_{t_0}; \phi))}{\exp(r(s_{t_0}, a_{t_0}; \phi)) + \exp(r(s_{t_1}, a_{t_1}; \phi))} \tag{4}$$

In this prediction model, trajectories that exhibit more ideal behaviors are considered to receive higher cumulative predicted rewards from the reward function $\hat{r}$. To align predicted rewards $\hat{r}$ with preference labels, in PbRL, the update of the reward function is treated as a binary classification problem. Specifically, with a dataset $D$ containing paired trajectories and their preference labels, the reward function is updated by minimizing the cross-entropy loss $L_{CE}$ between the preference prediction model and preference labels:

$$L_{CE}(\phi) = -\mathbb{E}_{\tau_0, \tau_1, y \sim D}[(1 - y)\log P(\tau_0 \succ \tau_1; \phi) + y \log P(\tau_1 \succ \tau_0; \phi)] \tag{5}$$

## 4 METHOD

In this section, we introduce a preference-based credit assignment method for episodic reinforcement learning. By introducing preferences for each trajectory, our method efficiently assigns credit-rewards at each time step, providing a guideline for policy learning even when the episodic reward is noisy. Specifically, our method evaluates trajectories qualitatively by introducing a global preference. For each trajectory, the contribution of each decision step to the final outcome is estimated and used to assign episodic rewards back to individual decisions. This approach achieves effective credit assignment and mitigates the adverse effects of noise in episodic rewards.

### 4.1 PREFERENCE MODELING

As it is difficult to distinguish trajectories with similar quality, inspired by (White et al., 2024), in our method, trajectory levels $C \in \{bad, \ neutral, \ good\}$ are constructed according to the episodic rewards and the entire trajectory space. We also refer to this as Global Preferences, which describes the overall quality of actions or states within a trajectory. Compared to the original PbRL settings, it can be seen that a "neutral" label is added as a preference label for easier comparison and more accurate evaluation of the trajectory quality. Then, with the observed dataset $D := \{(\tau_i, R_{ep}(\tau_i))\}_{i=1}^L$, also known as the trajectory space, where $L$ is the number of trajectories, and $R_{ep}(\tau_i)$ is the episodic reward of trajectory $\tau_i$, we first normalize $R_{ep}(\tau_i)$ according to the trajectory space $D$ to a relative trajectory level $z(R_{ep}) \in [0, 1]$:

$$z(R_{ep}) \triangleq \frac{R_{ep} - \min_{\tau' \sim D}\left(R_{ep}\left(\tau'\right)\right)}{\max_{\tau' \sim D}(R_{ep}(\tau')) - \min_{\tau' \sim D}(R_{ep}(\tau'))} \tag{6}$$

Next, based on $z(R_{\text{ep}})$, a global preference $y \in C$ is assigned to each trajectory as its label, and the global preferences are defined as follows:

$$y \triangleq \begin{cases} good, & z > \eta_2 \\ neutral, & \eta_1 \leq z \leq \eta_2 \\ bad, & z < \eta_1 \end{cases} \tag{7}$$

Where $\eta_1$ and $\eta_2$ are threshold hyperparameters for global preference classification that control the boundary of classification, which allow us to assign global preference to each trajectory simply by adjusting these hyperparameters.

## 4.2 PREFERENCE PREDICTION

As mentioned in Section 3, PbRL can be used to design the reward function in episodic reinforcement learning. However, the designed reward function only evaluates the quality of each trajectory, and the credit assignment within each trajectory is still not addressed. In our approach, we first show how to determine the relevance between each state-action transition and the trajectory preference. Then, credit can be assigned to past decision steps according to such relevance and the episodic rewards.

In detail, a new preference prediction model $P(\mathbf{p}_y | s_t, a_t, \tau_{1:t-1}; \varphi)$ with the incorporation of historical information $\tau_{1:t-1}$ is firstly designed, where $\varphi$ represents the optimization parameters, and $\mathbf{p}_y = [p_{\text{good}}, p_{\text{neutral}}, p_{\text{bad}}]$ is a set of probabilities that denotes the relevance between each state-action transition and the "bad", "neutral", and "good" preferences, respectively. Each probability ranges from $[0, 1]$ and satisfies $\sum \mathbf{p}_y = 1$. In this prediction model, by predicting the conditional probability distribution of the preference labels for each state-action transition, the relevance between each state-action transition and the trajectory preference is converted into a classification problem.

To realize the prediction model $P_\varphi$, a causal Transformer framework (Radford et al., 2018) (namely the Transformer network with causal masked self-attention) is adopted as the primary framework. Causal Transformer consist of stacked "masked self-attention layers". Each layer receive input tokens, and outputs preserve the same dimensions. These layers allow the model to take into account the information of all previous tokens in the sequence when processing each token. In our method, the token output of $s_t$ is ignored because the token output of $a_t$ has already taken into account all previous input tokens (including $s_t$). Therefore, when processing the state-action pair $(s, a)$, only the token output of each $a_t$ token needs to be retained to generate a prediction of the state-action. Besides, at each time step, the output of the model is only related to historical information, establishing the temporal relationships of the inputs, allowing the model to generate predictions for each transition step that incorporates historical information.

Specifically, as shown in Figure 1, the model takes a trajectory $\tau$ of length $T$ and generates $2T$ embeddings (for states and actions). These embeddings are learned through linear layers and added with shared positional encoding after normalization. Note that the same positional encoding is shared for state and action at each timestep. Subsequently, the outputs from the embedding layers are fed into the causal Transformer network, where they are processed by multiple linear layers and the Softmax

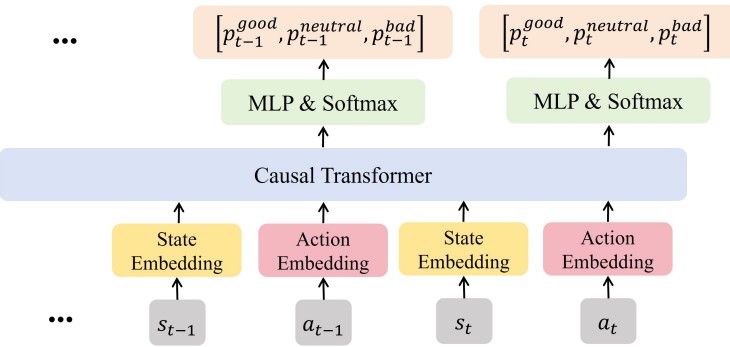

Figure 1: Overview of the prediction model structure.

function to produce the predicted probability of preference labels for each state-action transition $\left[ p_t^{\text{good}}, p_t^{\text{neutral}}, p_t^{\text{bad}} \right]_{t=1}^{T}$.

Then, for a trajectory, the predicted probability of preference labels $\mathbf{p}_y^{\text{traj}} = \left[ p_{\text{good}}^{\text{traj}}, p_{\text{neutral}}^{\text{traj}}, p_{\text{bad}}^{\text{traj}} \right]$ is defined as the mean of all transition prediction probabilities within the trajectory:

$$P_{\text{traj}}(\mathbf{p}_y^{\text{traj}}|\tau;\varphi) = \frac{1}{T} \sum_{t=1}^{T} P(\mathbf{p}_y|s_t, a_t, \tau_{1:t-1};\varphi) \tag{8}$$

Finally, for the training of the preference model, the cross-entropy loss between the aforementioned predicted results and the actual global preferences is utilized. As the distribution of different preference label trajectories is typically unbalanced, which might affect the accuracy of the loss function, we employ weighted sampling of different global preferences trajectories in the model training and incorporate a weight $w(x)$ based on the number of preferences into the loss function:

$$w(x) = \begin{cases} softmax \left( \frac{1}{\sum_{(\tau, \ y) \sim D} \mathbb{I}(y=x)} \right) & \sum_{(\tau, \ y) \sim D} \mathbb{I}(y = x) > 0 \\ 0 & \text{else} \end{cases} \tag{9}$$

Where $D$ is the trajectory space, $\mathbb{I}(\cdot)$ is the indicator function, $x$ and $y$ represent the preference label of trajectories. The weight $w(x)$ is set to be inversely proportional to the number of preferences in the trajectory space to balance the importance of different preferences in the loss function. Accordingly, the definition of the loss function is as follows:

$$L(\varphi) \triangleq -\mathbb{E}_{(\tau, \ y) \sim D} \left( \sum_{label \in C} w(label) \mathbb{I}(y = label) \log \left( P_{traj} \left( p_{label}^{traj} \Big| \tau;\varphi \right) \right) \right) \tag{10}$$

Note that to improve the sample efficiency of the model training, any sub-trajectory of a trajectory can be used as a training sample for the preference prediction model (see Appendix B).

## 4.3 CREDIT ASSIGNMENT

As in episodic reinforcement learning, the episodic reward provides the same or similar supervision to all decisions within a trajectory, which makes the learning of the RL agent slow and sample inefficient. In reality, the decisions at different time steps may have different impacts on the final performances. In our approach, by utilizing the proposed model $P_\varphi$, the probabilities of transitioning to different preferences can be predicted for each trajectory and at any time step. We define a Markov reward function $\hat{r}_t(\tau_{1:t}, (R_{\text{ep}}(\tau), y);\varphi)$ that can take any length trajectory $\tau_{1:t} = \{(s_k, a_k)\}_{k=1}^{t}$ as shown in Equation (11). The inputs of this reward function are the feedback information of the complete trajectory, i.e., the episodic reward and the corresponding preference:

$$\hat{r}_t(\tau_{1:t}, (R_{\text{ep}}(\tau), y);\varphi) \triangleq R_{\text{ep}}(\tau) \sum_{label \in \mathcal{C}} \mathbb{I}(y = label) P(p_{label}|\tau_{1:t};\varphi) \tag{11}$$

With such a designed reward function, the credit assignment of the episodic reward to past decision steps can be realized, and a Markov reward for any state-action transition at any time $t$ in the trajectory is provided. Consequently, the optimization goal of the policy is defined as:

Furthermore, thanks to the prediction model $P_\varphi$'s ability to process trajectories of any length, existing experience replay techniques can be used to update the policy. In the training, the predicted rewards are combined with the policy optimization to iteratively optimize both the reward model and the policy. The pseudocode of our algorithm is shown in Algorithm 1.

$$J(\theta) \triangleq \mathbb{E}_{\pi_\theta}[R(\tau)] = \mathbb{E}_{\pi_\theta} \left[ \sum_{t=1}^{T} \hat{r}_t(\tau_{1:t}, (R_{\text{ep}}(\tau), y);\varphi) \right] \tag{12}$$

---

**Algorithm 1:** Trajectory Preference-based Credit Assignment

---

**Input:** Input parameters
**Output:** Output results
**if** *online setting* **then**
    | Initialize $D \leftarrow \emptyset$.;
**if** *offline setting* **then**
    | Initialize $D$ from Dataset.;
    | Compute and store global preference labels $\{y_j\}_{j=1}^L$ for all trajectories from $D$ by (6)(7).;
**for** $\ell = 1, 2, \cdots$ **do**
    **if** *online setting* **then**
        Collect $K$ trajectories $\{\tau_j\}_{j=1}^K$ using the current policy.;
        Compute global preference labels $\{y_i\}_1^K$ for these trajectories by (6)(7).;
        Store trajectories $\{\tau_j\}_{j=1}^K$ and feedbacks $\{R_{ep}(\tau_j), y_j\}_{j=1}^K$ into the replay buffer
        $D \leftarrow D \cup \{\{(\tau_j, R_{ep}(\tau_j), y_j)\}_{j=1}^K\}$.;
    **for** $i = 1, 2, \cdots$ **do**
        Sample $H$ trajectories $\{\tau_j \in D$ weighted by $w(x)\}_{j=1}^H$ from replay buffer.;
        Sample subsequences $\{\sigma_j = \tau_{j,1:n_j}\}_{j=1}^H$ where $n_j \in \{1, 2, \ldots, T\}$ from these
         trajectories.;
        Estimate preference prediction loss $L(\varphi)$ using sampled subsequences by (10).;
        Perform a gradient update on the reward model $\hat{r}_t$:;

$$\varphi \leftarrow \varphi - \alpha \nabla_\varphi L(\varphi),$$

        where $\alpha$ denotes the learning rate.;
        Sample $N$ transitions with their history and trajectory feedbacks from replay buffer:

$$\{(s, a, \tau_h, R_{ep}(\tau), y)_j \in D\}_{j=1}^N,$$

        where $\tau_h$ denotes the trajectory before transition $(s, a)$.;
        Perform policy optimization using sampled transitions and the learned reward function
        $\hat{r}_t$.;

---

# 5 EXPERIMENTS

In the experiments, we first show that current credit assignment methods are susceptible to noised rewards while our proposed approach is more robust. Subsequently, to further validate our method's performance, it was tested and compared with SOTA baselines on the MIMIC-III dataset, which contains information on 17,898 sepsis patients. Finally, we briefly discuss and analyze the reliability and reasonability of the actions produced with the guideline of the credit assigned rewards. Besides, the addtional experiments can be see in Appendix E.

## 5.1 A NAVIGATION MAZE BENCHMARK TASK

**Setup** One of MuJoCo's benchmark tasks, namely PointMaze is first employed as the testbed, where the task is to s control a 2-DoF sphere to reach a goal in a closed maze, and the episodic reward is +1 once the task is successfully completed, while the failure of the task results in a negative reward equaling to the negative distance between the end position and the goal. Such a reward setting, though imprecise, can still reflect the trajectories' quality. (More details of this environment are given in Appendix C).

**Baselines** It is worth noting that we did not use the PbRL method as a baseline because our method aims to solve credit assignment problem in delayed and noised reward environments, which differs in focus from PbRL. More importantly, most PbRL methods are based on specific human preference datasets and cannot be applied in online environments or offline datasets with numerical annotations.

Therefore, in this experiment, our method ($\eta_1 = 0.8$, $\eta_2 = 0.55$) was compared with credit assignment methods RRD (biased), RRD (unbiased) (Ren et al., 2022), IRCR (Gangwani et al., 2020), and the RL algorithm vanilla SAC (Haarnoja et al., 2018), as well as the SAC algorithm using a discount factor $\gamma$ ($\gamma = 0.99$). To ensure a fair comparison, the same hyperparameters as in the SAC algorithm were employed for all the algorithms.

**Results**  As can be seen from Figure 2, our method outperforms the baseline methods in terms of both the average accumulated rewards and the convergence speed. It is noted that RRD and IRCR, which are state-of-the-art (SOTA) credit assignment methods, perform even worse than the SAC algorithm with $\gamma$-return. The possible explanation is that existing credit assignment approaches are susceptible to noised rewards, and the noised rewards

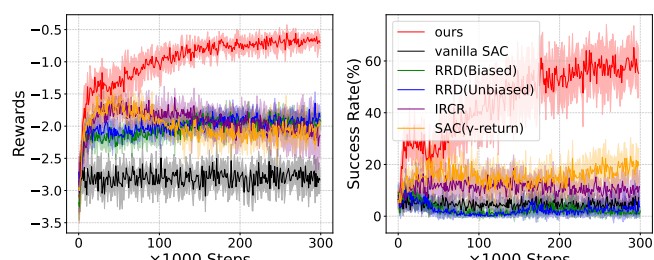

Figure 2: The learning curves for each algorithm on the PointMaze task with episodic rewards, displaying the average accumulated rewards and success rates for each algorithm over five random seeds, with the shaded area indicating the standard deviation.

will lead the learning of the agent in the wrong direction. In contrast, with the incorporation of PbRL and the preference-based credit assignment reward function, our approach can mitigate the adverse effects of noisy episodic rewards.

## 5.2 SEPSIS TREATMENT EXPERIMENTS

In the experiments, our method was verified on a subset of the MIMIC-III database (Johnson et al., 2016), which includes information about 17,898 sepsis patients. The goal of the task is to learn a treatment strategy by controlling the dose of intravenous infusion and vasopressor (please refer to Appendix D for more details of the sepsis treatment problem).

### 5.2.1 SETUP

**Evaluation metrics**  As directly deploying online evaluation models in clinical settings is risky, the performances of the policies are evaluated by using the Weighted Doubly Robust (WDR) off-policy estimator (Raghu et al., 2018a), where WDR is the estimated expected return of the offline policy and is a quantification of the treatment effect, i.e., the higher the WDR value, the better the treatment effect. Additionally, following the evaluation setup in (Liang et al., 2023), we calculated the Estimator Mortality for 30 sub-intervals over the expected return interval $[-15, 15]$ to estimate the patient's mortality of the trained policy, and the relationship between expected returns and Estimator Mortality is provided in Appendix D.3.

**Baselines**  We select the current state-of-the-art algorithms, the improved D3QN based on episodic control (Liang et al., 2023) (referred to as D3QN-EC in this paper) and some other advanced baselines including Mix of Expert (MoE) (Peng et al., 2018), D3QN-H and D3QN-A based on D3QN (Raghu et al., 2017; Raghu, 2019), Conservative Q-Learning (CQL) (Kaushik et al., 2022), and Model-based algorithms based on PPO (Raghu et al., 2018b). Some Q-table algorithms, such as Tabular Q-Learning (TQL), TQL-History (Tang et al., 2020), and Fitted Q-Iteration with Random Forest (FQIRF) (Raghu et al., 2018a), are also compared and discussed.

**Reward setting**  To verify the performance of the proposed method, the final reward provided at each trajectory in the dataset is used as the episodic reward to train our credit assignment prediction model. The original reward existing in each step of the trajectory will then be replaced by the predicted reward given by our model. For the policy learning, the current state-of-the-art algorithm

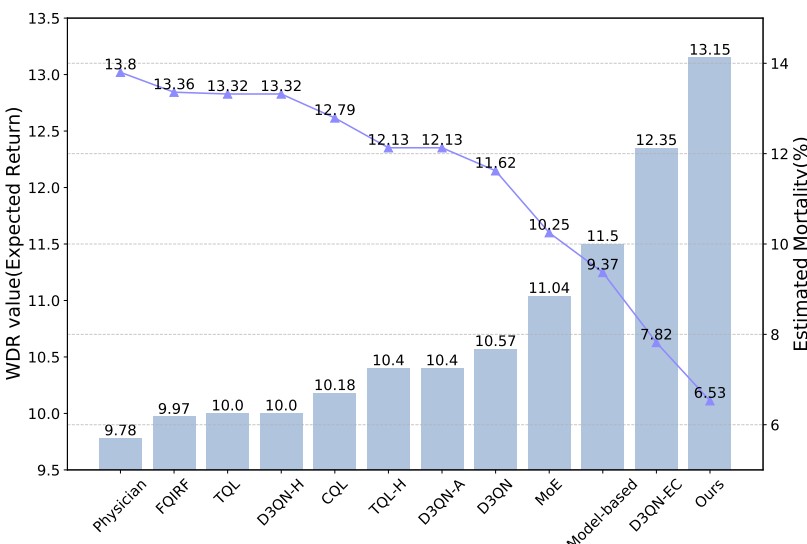

Figure 3: The WDR estimator results and Estimator Mortality for our proposed method and all baselines. The results show the average over 10 runs.

D3QN-EC is adopted for training and evaluation. For baselines, besides the final reward, the rewards at each time step $r_{\text{baseline}}(s_t, a_t)$ during the treatment period are also provided, which is manually designed according to expert knowledge (see the Appendix D.4 for more details).

### 5.2.2 RESULTS

As can be seen from Figure 3, compared to the baselines, the overall evaluation shows a significant performance improvement in our algorithm. Specifically, our algorithm's estimated WDR is $13.15 \pm 0.68$, nearly 6.5% higher than the D3QN-EC algorithm ($12.35 \pm 0.58$), and 34.5% higher than the physician (data) strategy. In terms of mortality prediction, the learned policy can reduce the patients' mortality to 6.53%, also outperforming other baselines.

To further verify the effectiveness of our method in dealing with noisy and delayed feedback, we add different random noise to the episodic rewards of each trajectory in the training set, which increases the uncertainty of the episodic reward and is defined as $R'_{ep} = R_{ep} + \xi_n$, where $\xi_n \sim \text{Uniform}(-n, n), n \in \{1, 3, 5, 10\}$ represents an unbiased sampling distribution from $-n$ to $n$. In the training process, the noised rewards were normalized and scaled to the same range $[-15, 15]$ as the dataset. Table 1 shows that the WDR estimator results for D3QN-EC with our method under different noised

Table 1: Results under different noised rewards.

| Noise $n$ | WDR Value |
|-----------|-----------|
| 0 | $13.15 \pm 0.68$ |
| 1 | $12.87 \pm 0.72$ |
| 3 | $12.54 \pm 0.91$ |
| 5 | $11.86 \pm 1.24$ |
| 10 | $11.58 \pm 1.86$ |

rewards. From the experimental results, we can see that the impact of noise on policy performance shows a linear relationship, i.e., as the noise increases, the performance of the policy gradually deteriorates and becomes unstable. Nevertheless, our method still demonstrates a performance advantage even under the setting of $n = 10$, outperforming the model-based method shown in Figure 3. This suggests to some extent that our method can effectively mitigate the adverse effects of noise.

Finally, we visualized the learned strategies of our method, the best baseline (D3QN-EC) and physician in Figure 4, and we can see that both the methods using manually designed rewards and our approach reduce the no-medication actions, providing clinicians with a richer set of medication recommendations. However, our method's action distribution more closely aligns with the physicians' strategies, particularly in the use of vasopressors. While D3QN-EC tends to recommend high doses

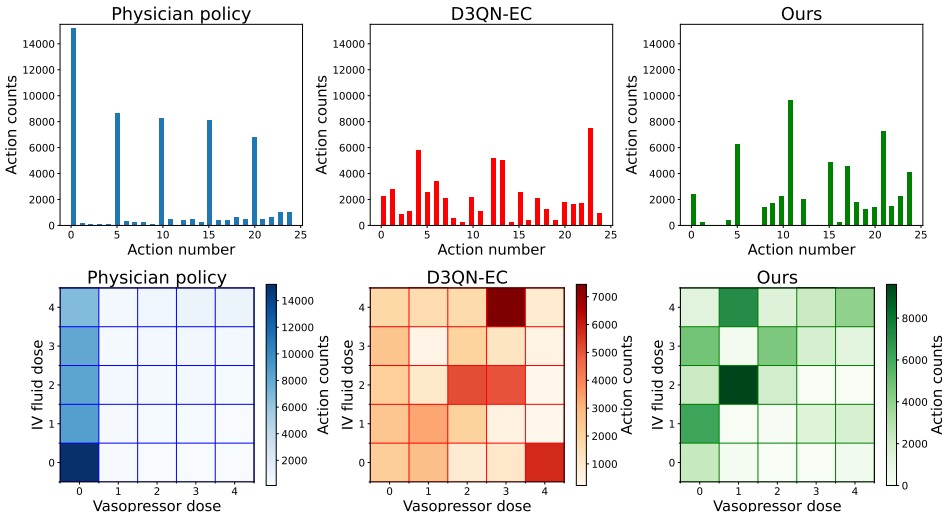

Figure 4: Action Distributions for physician (blue), D3QN-EC (red) and our methods (green) over test set. Upper: Bar chart of action distribution, with the x-axis representing actions and the y-axis representing the number of actions. Lower: Heatmap of different drug usage combinations (actions), with larger numbers indicating higher dosages. The bar on the right side of each chart represents the color gradient corresponding to the quantity.

of vasopressors, both the physicians' strategy and our method take a conservative approach to their use. Moreover, our strategy is also similar to the sepsis survivor campaign guidelines (Evans et al., 2021) in the use of intravenous fluids where intravenous fluids are recommended for sepsis patients.

## 6 CONCLUSION

In this paper, to address the issue of existing credit assignment methods struggling to achieve effective assignment in noisy episodic rewards, we propose a trajectory preference-based credit assignment method by leveraging the benefits of PbRL. Compared to existing credit assignment methods, our method provides a new form of relative feedback by constructing global preference feedback for trajectories, alleviating the assignment error caused by noisy episodic rewards. Additionally, the designed prediction-based reward function provides dense rewards to agents during policy learning, improving the learning efficiency. In experiments on the MuJoCo-based PointMaze online environment and the sepsis treatment dataset, we verified the adverse effects of episodic reward noise and demonstrated the effectiveness and applicability of our method. The results show that our method has the capability of credit assignment even when rewards contain obvious noise. In the sepsis treatment, it is worth noting that our method can guide agents to learn better and more realistic behavioral strategies than other baselines, which may provide more effective treatment suggestions for clinicians.

**Limitations and future works** From our method, we can also see that it still has some limitations when extends into practical applications, such as the need for adjusting numerous parameters and the high time cost facing long trajectory lengths. We leave these issues as part of future work, hoping to generate more interpretable guidance rewards using simple techniques with less parameters and more effective models.

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

## A  SOLUTIONS OF DELAYED REWARDS

In reinforcement learning settings, agents are usually assumed to receive dense real-time rewards as feedbacks during interactions with the environment. However, in many real-world applications, for example, medical decision-making and autonomous driving, such dense rewards are not available (Efroni et al., 2021), and reward signals are delayed and sparse, where in extreme cases the rewards can only be received at the end of a trajectory, i.e., episodic reinforcement learning. The delayed and sparse rewards pose great challenges to the credit assignment of RL agents. In the literature, potential-based reward shaping methods have been proposed to provide rapid credit assignment while ensuring optimal strategies (Ng, 1999), but constructing a potential function for each state often requires rich expert knowledge of the domain. While evolution-based algorithms (Zou et al., 2021) perform gradient-free optimization directly in the trajectory policy space, making them insensitive to delayed rewards, the ignoring of temporal information makes these methods less efficient than typical RL algorithms (Gangwani et al., 2020). Recent work has attempted to apply Transformers to sequential decision-making (Chen et al., 2021), which enables agents to be effective in tasks with delayed and sparse rewards, but ample data is required in the offline pre-training stage of these approaches to acquire reliable policy models.

## B  TRAINNING WITH SUB-TRAJECTORY

In our method, to improve sample efficiency, we follow the assumptions of RRD (Ren et al., 2022) and use random subsequences of trajectories to train the reward model $\hat{r}_t(\tau_{1:t}, (R_{ep}(\tau), y); \varphi)$ and approximate the episodic return of sub-trajectory $R_{ep}(\tau_{1:t})$ to reduce computational costs:

$$R_{ep}(\tau_{1:t}) \approx \mathbb{E}_{\sigma \sim \rho_T} \left[ \frac{|\mathcal{I}|}{T} R_{ep}(\tau) \right] \approx \frac{|\mathcal{I}|}{T} R_{ep}(\tau) \tag{13}$$

where $\mathcal{I}$ represents a subset of trajectory indices, $\rho_T = \text{Uniform}(\mathcal{I} \subseteq \{1, \ldots, T\} : |\mathcal{I}| = k)$ represents an unbiased sampling distribution, and $k$ is the number of sets $\mathcal{I}$. In this structure, the predictive reward function can approximate the environment's trajectory returns from a small subset of state-action pairs, allowing the algorithm to be trained via minibatch sampling.

## C  THE NAVIGATION MAZE BENCHMARK TASK

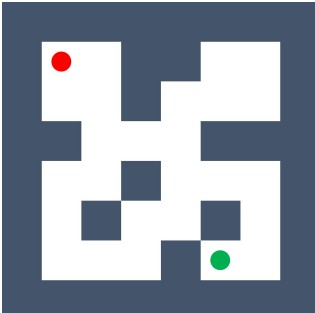

Figure 5: PointMaze (Medium)

### C.1  ENVIRONMENT

As illustrated in figure 5 the task involves controlling a 2-DoF sphere (the red sphere) driven by forces in the Cartesian x and y directions to reach a goal (the green sphere) within a closed maze. To increase the complexity of the scenario, the positions of both the sphere and the goal are randomized at the start of each round. The agent's state and actions are set as continuous inputs, representing position, target information, and the magnitude of linear force applied on the controlled sphere in the x and y directions (i.e., force generated along a straight line on the object). Agents only receive episodic rewards at the end of the trajectory, with the maximum trajectory length set to 100 steps.

## C.2 DETAILS OF IMPLEMENTATION

For all algorithms of credit assinment we use SAC (Haarnoja et al., 2018) as basis and Adam as all methods' policy optimizer. Hyperparameters which are common to all methods are shown in Table 2. All algorithms were trained for 30 million environment steps. We did not anneal learning rates for any of the methods during training, since we found this yielded similar or better performance and simplified the setup. The experiments were conducted on an Ubuntu system equipped with two Intel xeon Platinum 8373C (36 core and 2.6Ghz server processor), a Nvidia 4090 24Gb graphics and 128Gb Memory.

Table 2: Hyperparameters for PointMaze.

| Hyperparameter | Value |
|---|---|
| **Common Hyperparameters** | |
| Policy learning rate | 0.001 |
| Smoothing coefficient | 0.005 |
| Temperature parameter | 0.2 |
| Reward/Prediction Model learning rate | 0.001 |
| Warmup steps | 10000 |
| Replaybuffer size | 1000000 |
| Batch size | 256 |
| Hidden dimension of SAC | 64 |
| **Causal Transformer Model of Our Method** | |
| Embedding dimension | 32 |
| Encoder hidden layer size | 128 |
| Number of attention heads | 2 |
| Number of layers | 3 |
| Dropout rate(embedding, attention, residual connection) | 0.1 |
| Model learning rate | 0.001 |

# D SEPSIS TREATMENT

## D.1 DATASET

In the experiments, our method was verified on a sub-dataset of the medical MIMIC-III database (Johnson et al., 2016), which includes information about 17,898 sepsis patients. The dataset records static features (e.g. demographic), past treatment history, a summary of hourly observation (mean, maximum, and minimum within an hour) of all laboratory values within patients' first 72-hour ICU stay and information on intravenous fluid and vasopressor doses of patients during treatment.Table 3 shows the basic information and Table 4 shows the detailed features of the dataset.

Table 3: Summary statistics for the patient cohort.

| | % Female | Mean Age | Hours in ICU | Total Population |
|---|---|---|---|---|
| **Survivors** | 43.6 | 63.2 | 57.6 | 15,583 |
| **Non-Survivors** | 47.0 | 69.9 | 58.8 | 2,315 |

## D.2 DATAPROCESSING

To facilitate model training, we followed Raghu et al.'s (Raghu et al., 2017) preprocessing operations on the dataset, which included handling missing values, denoising, clipping, and normalization. The dataset was then split using 80% for training and validation and 20% for testing. Moreover, each sepsis patient's hospitalization data forms a trajectory in the form of a $\langle s, a, r, s', done \rangle$ tuple with a time step of 4 hours, where the state $s$ represents the patient's 48 physiological indicators, action $a$ represents the 25 dosing regimens of the two drugs, reward $r \in [-15, +15]$ quantifies the

Table 4: The physiological features of sepsis treatment dataset.

| **Vital Sign/Laboratory** | | | | |
|---|---|---|---|---|
| Glasgow Coma Scale | Heart Rate | Sys. BP | Dia. BP | Mean BP |
| Respiratory Rate | Body Temp (C) | FiO2 | Potassium | Sodium |
| Chloride | Glucose | INR | Magnesium | Calcium |
| Hemoglobin | White Blood Cells | Platelets | PTT | PT |
| Arterial pH | Lactate | PaO2 | PaCO2 | PaO2 / FiO2 |
| Bicarbonate (HCO3) | SpO2 | BUN | Creatinine | SGOT |
| SGPT | Bilirubin | Base Excess | | |
| **Demographics/Static** | | | | |
| Readmission status | Gender | Weight | Ventilation | Age |

fluctuation in the patient's condition and is given at the end of the trajectory, and flag done $\in \{1, 0\}$ indicates whether the patient survived at the end of the treatment trajectory. It should be noted that $s$ is the average of the physiological indicators over the 4-hour period, and $a$ is the maximum dose administered during the 4-hour treatment period.

### D.3 EVALUATION METRICS

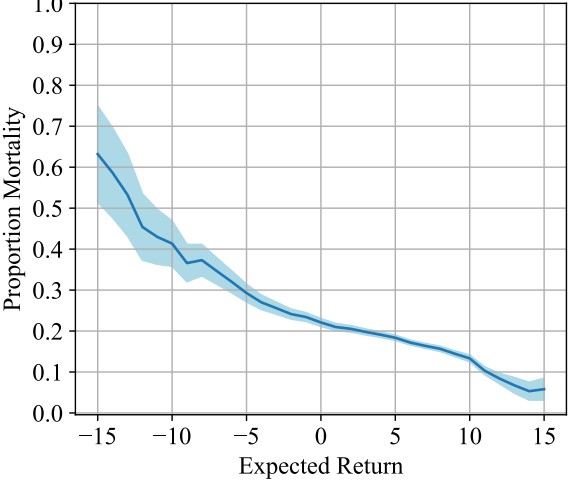

Figure 6: The relationship between expected returns and estimated mortality.

The Weighted Doubly Robust (WDR) estimator extends the Importance Sampling (IS) estimator by introducing a value bias term and a sampling weight factor $\omega$ to ensure low variance properties. This can be expressed as:

$$\text{WDR}\left(D_{\text{test}}\right) := \sum_{i=1}^{N} \sum_{t=1}^{T} \gamma^{t-1} \omega_t^i r_t^{H_i} - \sum_{i=1}^{N} \sum_{t=1}^{T} \gamma^{t-1} \left( \omega_t^i \hat{Q}^{\pi_e}\left(S_t^{H_i}, A_t^{H_i}\right) - \omega_{t-1}^i \hat{V}^{\pi_e}\left(S_t^{H_i}\right) \right)$$

$$(14)$$

Where the $D_{\text{test}}$, $N$, $T$, and $H_i$ represent the test dataset, the number of patients, the time steps of each treatment trajectory, and the ICU-stay state trajectory of the $i$-th patient, respectively. Following the suggestions of Raghu et al. (Raghu et al., 2018a), we obtain the key parameters in WDR: $\hat{Q}^{\pi_e}(s, a)$, which is the action-value function under policy $\pi_e$ and also the evaluation target; the

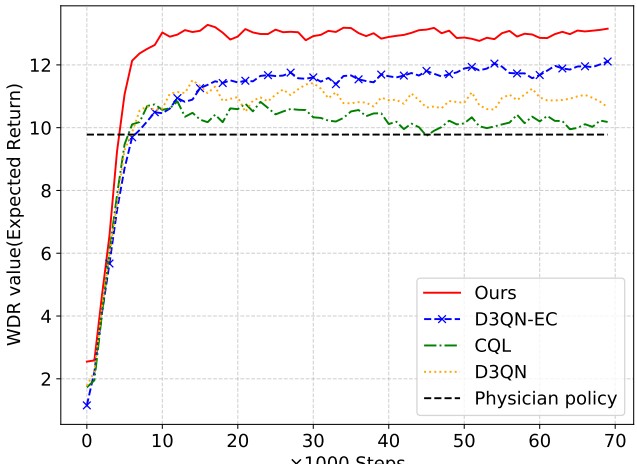

Figure 7: The average WDR value training curves for our method and selected baselines, each curve obtained by training strategies initialized from 10 random seeds.

caculation of state value $\hat{V}^{\pi_e}(s)$ is $\sum_a \pi_e(a|s)\hat{Q}^{\pi_e}(s,a)$, where the corresponding policy $\pi_e$ is obtained from $softmax\left(\hat{Q}(s,a)\right)$. Additionally, the data policy $\pi_b(a|s)$ involving sampling weights $\omega_t^i \propto \frac{\pi_e(a|s)}{\pi_b(a|s)}$ of policy distribution is a physician policy obtained using KNN approximation from real data. Specifically, we caculate the distance from a given state $s^*$ to all states in state space and count the number of states the nearest $k = 300$ states that select action $a$, i.e., $N(a|s^*)$. Then, the action probability $\pi_b(a|s^*)$ is approximated as the proportion $\frac{N(a|s^*)}{K}$.

In the section 5, the Estimator Mortality of models in Figure 3 were obtained from Figure 6 by utilizing the estimated WDR value, i.e., interpolated from the standard Expect Return-Proportion Mortality correspondence (Raghu et al., 2017). Figure 6 is derived by separately counting the real mortality for each of the 30 subintervals over expected return interval $[-15, 15]$, where the expected return corresponds to the real discounted return of the patient trajectories.

## D.4 REWARDS WITH EXPERT KNOWLEDGE

In the MIMIC-III dataset, the final reward is the only objective reward, which is given based on the patient's final actual treatment outcome. However, for the baselines, different from our approach where only the episodic reward is required, following other work in the literature, besides the final reward, the rewards at each time step $r_{\text{baseline}}(s_t, a_t)$ during the treatment period are also provided. These rewards are manually designed and defined as follows:

$$
\begin{aligned}
r_{baseline}(s_t, a_t) = & \ C_0 \mathbb{I}\left(s_{t+1}^{SOFA} = s_t^{SOFA} \text{ and } s_{t+1}^{SOFA} > 0\right) \\
& + \ C_1\left(s_{t+1}^{SOFA} - s_t^{SOFA}\right) \\
& + \ C_2 \tanh\left(s_{t+1}^{Lactate} - s_t^{Lactate}\right)
\end{aligned}
\tag{15}
$$

Where the "SOFA" (Sequential Organ Failure Assessment) outlines the extent of a patient's organ failure and the "Lactate" is a measure of cell hypoxia that is higher in septic patients; Following the work of Liang et al. (Liang et al., 2023), the hyperparameters are set to $C_0 = -0.025, C_1 = 0.125, C_2 = -2$.

## D.5 Additional details about experiment

The curves in Figure 7 correspond to Figure 3, which is according to WDR values and shows that strategies using predicted rewards achieve higher WDR value and faster convergence compared to those using manually designed rewards, as predicted rewards provide richer task information.

## D.6 Details of implementations

The hyperparameters of our method and policy learning are shown in Table 5, which gives our method, policy learning (common to all baselines) and some parameters unique to other baselines. Note that to ensure optimal performance of baselines, the MoE and Model-based shown in Figure 3 are the results of original works. The computer resources during experiments are the same as in the PointMaze experiments.

Table 5: Hyperparameters for Sepsis Treatment.

| Hyperparameter | Value |
| --- | --- |
| **Our Method** | |
| Optimizer | Adam |
| Learning rate | 0.001 |
| Embedding dimension | 32 |
| Encoder hidden layer size | 128 |
| Number of attention heads | 2 |
| Number of layers | 3 |
| Dropout rate(embedding, attention, residual connection) | 0.1 |
| Training steps | 30000 |
| Model learning rate | 0.001 |
| **Policy Learning** | |
| Optimizer | Adam |
| Coefficients of importance weight $(\alpha, \beta, \epsilon)$ | (0.6, 0.9, 0.01) |
| Discount factor $\gamma$ | 2 |
| Training steps | 70000 |
| Update rate $\tau$ | 0.001 |
| Regularisation Coefficient $\lambda$ | 5 |
| Learning rate | 0.0001 |
| Hidden layer size | 128 |
| **D3QN-EC** | |
| Tunning factor of D3QN-EC $\mu$ | 0.3 |
| **CQL** | |
| Update Coefficient of CQL $\alpha$ | 0.003 |
| **FQIRF** | |
| The number of trees | 80 |

# E Additional Experiment

## E.1 Additional experiments on the adverse effects of reward noise

We conducted additional experiments in PointMaze to further help readers understand the reward noise problem and its impact. In order to improve the accuracy of the final feedback, the experiment refined the composition of the episodic reward, giving the distance from the target point at each time step as a reward, and accumulating it to the last step (the original setting only uses the distance from the target point at the last time step as a reward).

The experimental results in the Table 6 show that RRD(biased) and RRD(unbiased) perform much better than the original reward setting in the delayed cumulative reward setting, indicating the adverse

effects of environmental feedback noise. Compared with the performance of our method in the one-step episodic reward setting (-0.68±0.12), it also shows that our method can be effective in an environment with obvious noised rewards at the expense of certain performance.

Table 6: The results in different reward setting

| Metric | RRD (unbiased) | RRD (biased) |
|---|---|---|
| Rewards (one-step episodic reward) | -1.81 ± 0.13 | -1.88 ± 0.17 |
| Rewards (multi-steps accumulated episodic reward) | -0.56 ± 0.11 | -0.64 ± 0.16 |

### E.2 THE ABLATION EXPERIMENT ABOUT HYPERPARAMETER

The table 7 below shows the comparison result of a set of ablation experiments with different parameters. We conducted at least 4 repeated experiments, in which each experiment is trained for 70,000 steps, and finally took the average results of the at least 4 seeds for comparison.

Preliminary results indicate that the model performs best when $(\eta_1, \eta_2) = (0.3, 0.7)$ (the hyperparameter setting in the sepsis experiment). It is noteworthy that regardless of whether the range of $(\eta_1, \eta_2)$ approaches equality or the maximum difference, the model's performance deviates from the optimal value.

Table 7: WDR Values under Different Hyperparameters.

| $(\eta_1, \eta_2)$ | (0.4, 0.6) | (0.33, 0.66) | (0.3, 0.7) | (0.2, 0.8) | (0.1, 0.9) |
|---|---|---|---|---|---|
| WDR Value | 12.72 ± 0.41 | 12.81 ± 0.62 | **13.15 ± 0.68** | 12.22 ± 0.79 | 12.08 ± 0.61 |

### E.3 A 2D GRID-WORLD BENCHMARK: MINIGRID

We conducted experiments on several benchmark tasks in Minigrid that meet the credit assignment requirements. The description of the benchmark tasks is as follows:

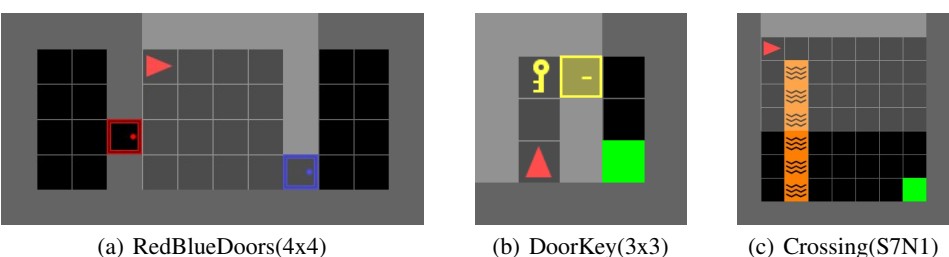

(a) RedBlueDoors(4x4)  (b) DoorKey(3x3)  (c) Crossing(S7N1)

Figure 8: Minigrid Benchmark

**RedBlueDoors(8x8)**: The agent is randomly placed within a room with one red and one blue door facing opposite directions. The agent has to open the red door and then open the blue door, in that order. (The size of the map is 8x8).

**DoorKey(8x8)**: This environment has a key that the agent must pick up in order to unlock a door and then get to the goal square. (The size of the map is 8x8).

**SimpleCrossing(S11N5)**: The agent has to reach the goal square on the other corner of the room while avoiding rivers of wall. (S: size of the map SxS. N: number of valid crossings across walls from the starting position to the goal.).

In the experiment, we selected representative baselines and our method for testing, and set the rewards of all tasks to episodic rewards. At the end of the trajectory, a reward of $r = 1 - 0.9 *$ $(step_{count}/step_{max})$ is given for success, and $r = 0$ for failure. In addition, the maximum step size

of the tasks is set to 640 (RedBlueDoors), 640 (DoorKey), and 484 (SimpleCrossing). It should also be noted that since the environment action space is discrete, we selected PPO as the new benchmark algorithm (the original benchmark algorithm SAC is suitable for continuous action space). Experimental results show that our method is still better than the SOTA credit assignment method.

Table 8: Comparison of different methods across tasks.

| Task | PPO | RRD (unbiased) | Ours |
|------|-----|----------------|------|
| RedBlueDoors (8x8) | 0.009±0.005 (5M steps) | 0.035±0.012 (5M steps) | **0.901±0.011 (5M steps)** |
| DoorKey (8x8) | 0.118±0.096 (3.5M steps) | 0.134±0.082 (1.5M steps) **0.942±0.010 (3.5M steps)** | **0.946±0.006 (1.5M steps)** |
| SimpleCrossing (S11N5) | 0.254±0.134 (10M steps) | 0.388±0.192 (10M steps) | **0.930±0.067 (10M steps)** |

## F  DISCUSSION ON THE RELATIONSHIP BETWEEN OUR METHOD AND PBRL

According to the description of Wirth et al.(Wirth et al., 2017), the problem of learning a preference model $C(a \succ a'|s)$ can be phrased as a classification problem trying to correctly predict all observed preferences. The pairwise preference predictions of the trained classifiers $C_{ij}$ may be combined via voting or weighted voting, where each prediction issues a vote for its preferred action. The resulting count $k(s, a)$ for each action $a$ in state $s$ correlates with $\rho((s, a_i) \succ (s, a_j)|s)$ if the classifiers $C$ approximate $\rho$ well:

$$k(s, a) = \sum_{\text{for all } a_i \in A(s), a_j \neq a} C(a_i \succ a_j|s) = \sum_{\text{for all } a_i \in A(s), a_j \neq a} C_{ij}(s)$$

Then, the optimal action can be defined as $a^* = \text{argmax}_{a'} k(s, a')$.

Obviously, our method is similar to the above definition, and both output the probability of the corresponding preference for each state-action pair $(s, a)$ (corresponding to the number $k(s, a)$), i.e., $\mathbf{p}_y$. Under this premise, the optimal action can be described as $a^*_{\text{ours}} = \text{argmax}_{a'}(\mathbf{p}_{y=\text{good}}|s, a', \tau_{1:s_{t-1}})$.

However, the focus of our work is on credit assignment, which requires measuring the impact of each state-action pair $(s, a)$ on the final result $y'$. Our approach is to use the probability $(\mathbf{p}_{y=y'}|s, a, \tau_{1:s_{t-1}})$ of the preference corresponding to each $(s, a)$ and the final result $y'$ as the basis for credit allocation. In other words, our method uses the preference probability to obtain the correlation between each step and the final result to obtain dense rewards.

Therefore, it can be said that our method is preference-based, utilizing the concept of relevance in PbRL to guide the long-term credit assignment task.

