# OpenReview forum: "Preference-based Credit Assignment for Reinforcement Learning with Delayed and Noised Rewards"
_ICLR.cc/2025/Conference — ICLR 2025 Conference Withdrawn Submission_

### Official Review · Reviewer_9wJB · 2024-10-29

**Soundness:** 1
**Presentation:** 3
**Contribution:** 2
**Rating:** 3
**Confidence:** 3

**Summary:**

The paper addresses the episodic reinforcement learning (RL) setting, where a reward is only given at the end of an episode. Since actions taken before the final step do not provide any reward signal, learning a policy in this setting is challenging. To tackle this issue, the authors propose a novel method for credit assignment of the total episode return across all state-action pairs in the trajectory, creating a dense reward function that offers feedback after each action. The approach can be summarized at a high level as follows:

1. Assign the full trajectory a label indicating that it is either good, bad, or neutral. This label is generated by comparing the given trajectory to all other collected trajectories, and is referred to as a global preference by the authors.
2. Use a Causal Transformer to learn the label for every transition within the trajectory. The predicted probability of label for a trajectory is the mean over the predicted probability that each transition in the trajectory has that label. Therefore the learning objective is to minimize the (weighted) cross entropy loss between the global preference from step 1 and the predicted probability of that label. This results in a Causal Transformer that can take in a trajectory segment of any length as an input, and output the predicted label distribution for the final state-action pair in that segment.
3. Use the Causal Transformer trained in step 2 to construct a dense reward function. This reward function takes in a trajectory segment, the episode return for the full trajectory, and the global preference for the full trajectory. The reward for the last transition in the trajectory segment is then equal to the episode return times the predicted probability—outputted by the Causal Transformer—that the transition label is equal to the global preference.
4. The dense reward function from step 3 can then be used to learn a policy using any RL algorithm.
The authors outperform SOTA credit-assignment approaches in an episodic Mujoco task, and outperform all baselines in a suite of sepsis treatment experiments. The authors also note that their approach is more robust to a noisy reward signal, particularly in the sepsis treatment experiments.

I currently recommend rejecting this paper due to limitations in the proposed methodology, but I would be happy to amend my recommendation if the authors can clarify any misunderstandings. My hesitations related to the proposed methodology are outlined in the Questions section.

**Strengths:**

The paper clearly defines and justifies the problem they are addressing. Their proposed methodology is clearly described and easy to grasp. To the best of my knowledge, the proposed approach is also novel, interestingly bringing ideas over from the field of preference based RL to credit assignment. The authors also appear to achieve strong results, particularly when evaluating their approach in the Sepsis Treatment Experiments. They show how their approach is more robust to a noisy reward signal, and that its learned action distribution more closely aligns with the Physicians strategy; I appreciated this extra dimension of analysis. For me, clarity and empirical analysis serve as this papers key strengths.

**Weaknesses:**

My main concern is the technical soundness of the authors' proposed credit assignment approach. In the Questions section of this form, I list specific issues that I hope the authors can address to clarify any misunderstandings and potentially resolve my perceived concerns about their approach. If these questions are satisfactorily addressed, I would be willing to raise my soundness and contribution scores.

Another weakness of the paper revolves around the results in Section 5.1. The authors write “that RRD and IRCR, which are state-of-the-art (SOTA) credit assignment methods, perform even worse than the SAC algorithm with γ-return.” They then very briefly describe a possible explanation, but I would like to see a more rigorous explanation here. The SOTA methods for credit assignment are underperforming methods that perform no credit assignment; this seems like a pretty important finding that warrants further investigation. What about this specific environment or expirement setup makes it so that the authors are unable to reproduce the results of RRD and IRCR? Gangwani et al. show that IRCR vastly outperforms SAC with gamma=0.9 (similar to what the authors are calling gamma-SAC, except they use gamma=0.99) and Ren et al. shows that RRD outperforms IRCR, both in Mujoco tasks with episodic rewards. The authors should therefore provide a more detailed analysis of why RRD and IRCR underperform in their specific experimental setup. This may involve conducting additional experiments or analyses to isolate the factors causing this unexpected performance, or providing a more thorough comparison of their experimental conditions to those in the original RRD and IRCR papers. This would help clarify whether the issue is with the environment, the implementation, or some other factor.

**Questions:**

I have several key questions that center around Eq. 11. Addressing these questions will likely address my perceived weaknesses in the methodology of this paper. Firstly, the authors state that their designed reward function is Markov. But in Eq. 11, the reward at timestep t is dependent on the transitions from timesteps 1 to t. This seems to directly violate the Markov condition. Can you clarify how the reward function satisfies the Markov property given its apparent dependence on past transitions?

Additionally, with regards to Eq.11, say I have a full trajectory with 5 transitions [$sa_1$, $sa_2$, $sa_3$, $sa_4$, $sa_5$]. The transitions have the labels [neutral, neutral, neutral, good, good]. The trajectory has an episode return of 1 and a global preference of neutral. Assume I can correctly predict the preference label for any transition with p(label | .). Constructing a reward function with Eq. 11 would result in the following rewards [1, 1, 1, p(neutral | $sa_1$, $sa_2$, $sa_3$, $sa_4$) < 1, p(neutral | $sa_1$, $sa_2$, $sa_3$, $sa_4$, $sa_5$) < 1]. So the rewards assigned to the last two transitions, which are both labeled as “good”, are lower than the rewards assigned to the first three transitions, which are labeled as “neutral”. A similar argument would hold for why transitions labeled as “bad” might be assigned higher rewards than transitions labeled as “good”—particularly if the episode return is always positive. What am I misunderstanding here? Can you address this specific scenario and explain how the proposed method handles such cases where the assigned rewards seem counterintuitive to the transition labels?

---

### Official Review · Reviewer_KhdH · 2024-11-02

**Soundness:** 1
**Presentation:** 1
**Contribution:** 2
**Rating:** 3
**Confidence:** 3

**Summary:**

The paper proposes a method for preference-based credit assignment in reinforcement learning. The approach uses a model that classified trajectories into "good", "neutral", and "bad" based on the episodic reward. The authors employ a causal transformer to to predict the probability of "good", "neutral", and "bad" given a history of state-action pairs and use such learned preferences for credit assignment. The paper evaluates their method on two environments -- MuJoCo PointMaze and Sepsis Treatment data.

**Strengths:**

1. The paper tackles the important problem of credit assignment with sparse reward functions.
2. The categorization in to "good", "neutral", and "bad" might provide more fine-grained guidance compared to binary preferences.
3. The experimental results show that the method might be a promising avenue of research. In particular, the method seems to be robust to noise in the reward.

**Weaknesses:**

1. **Limited Experimental Validation.** The paper would benefit from more experiments. Currently, the authors only validate the method on two tasks, which might not provide enough evidence for the superiority of the method to previous ones.
2. **Trajectory Labeling.** The "good/neutral/bad" trajectory labels seem unnecessary given that the numerical reward are given at the end of the episode. The more fine-grained rewards might even be better (despite the noise).
3. **Notation Could be More Consistent and Clearer.** The paper seems to use different notation for the same thing in many places and the notation is sometimes confusing or not properly introduced.
	1. In Equation 3, it might be helpful to use $\tau_i, \tau_j$ to indicate trajectories rather than $A, B$ to be consistent with the rest of the paper.
	2. Line 187. It is unclear if $\hat{r}$ is the same as $r(\cdot, \cdot; \phi$).
	3. $C$ is defined as an element of a set in line 206, however $C$ seems to be the set itself in line 216.
	4. Line 241. It is unclear if $P_\varphi$ if the same as $P(\cdot \mid \cdot, \cdot, \cdot, \varphi)$
	5. In Equations 10 and 11, it would be helpful to use $x \in C$ rather than $\text{label} \in C$ to maintain consistency.
4. **Missing Summation and Indices in Equation 4.** Equation 4 should contain a summation over the time steps for each trajectory.
5. **Incorrect Claim on Markov Reward.** Line 307 claims that the learned reward function $\hat{r} (\tau_{{1:t}}, (\dots); \varphi)$ is Markovian, but this seems to be incorrect because of the dependence on the entire trajectory up to time step $t$.
6. **Confusing Definition of Episodic Reward.** It is unclear if the definition in line 161 implies that $r_t = 0, \forall t < T$ or if $r_t$ can be non-zero and the episodic reward is simply defined as the cumulative reward. Moreover, it is unclear what the difference in the two cases in Equation 1 is.
7. **The Claim in Lines 164-166 are not Supported.** It is not clear where it has been shown that rewards in episodic reinforcement learning are non-Markovian and that sample efficiency is worsened. Could the authors provide references to this or an explanation to clarify this point?
8. **Typo.** There is a typo in line 369. Should it say "is to control" rather than "is to s control"?
9. **Missing Content in Line 316.** The paragraph in line 316 ends with a column and seems to be missing something.
10. **Claim in Lines 67, 68 is not Accurate.** There exist methods for learning from preferences defined as positive and negative samples. Consider for example KTO for LLM finetuning.
11. **Claim in Line 230 is not Accurate.** Preference transformers does the credit assignment by learning a non-Markovian reward model.
12. **Figure 4 Can Be a bit Hard to Read.** The different color palettes and ranges for the heat maps can make Figure 4 harder to read. A suggestion is to use the same color scheme for all 3 settings and only have a single range for the action counts so that the 3 bottom plots are more easily comparable.
13. **Guaranteed Performance.** Line 88 states that the "performance of the agent can still be guaranteed." There is however no formal proof for the optimality of the policy with respect to the episodic reward.
14. **Reward Can be Used to Construct Pairwise Preferences.** In lines 133-135, it is stated that assigning pairwise preference labels is more expensive than setting a reward function. However, having access to the episodic reward to label data as good/neutral/bad, one can also easily construct pairwise preference at no human cost. Moreover, if one has access to the episodic reward to construct good/neutral/bad data, why would they not simply the episodic reward itself? This would be a more fine-grained learning signal.

**Questions:**

See **Weaknesses**. Additionally:
1. **How does the method compare with preference transformers?** It seems like PT can be trained online (see Appendix B "Implementation details of Hopper backflip") and should be compared with the proposed method. Moreover, some of the baselines in Section 5.2 are offline RL methods (e.g., CQL), so why not compare the method with PT as well?
2. **Why is it good to be close to the physicians' strategy if physicians perform the worst?** Figure 3 shows that the strategy employed by physicians is the worst, but in line 485, it seems like the authors claim that being close to the physicians' policy is positive. Could the authors clarify this discrepancy?
3. **How to Choose Thresholds.** It seems like the choice of thresholds $\eta_1, \eta_2$ would be quite important for the proposed method. How can one choose such parameters effectively? Why do the authors use $\eta_1 = 0.8, \eta_2 = 0.55$? It would be interesting to see an ablation on the choice of thresholds. What happens if $\eta_1 = \eta_2$, i.e., there are only "good" and "bad" trajectories but no "neutral" ones?
4. **Does Equation 11 Need The Full Trajectory?** The learned reward up to time step $t$ depends on the full trajectory $\tau$ via $R_{ep}(\tau)$. How does one use this during a policy interaction with the environment since $\tau_{t+1:T}$ is not realized yet? Does this mean that the reward is only compute at the end of a rollout?
5. **Unclear Reference in Line 110.** Is it supposed to be "Christiano et al." or "Ouyang et al." before "demonstrate that preference-based ..."?
6. **Why can't most PbRL methods be applied online?** As previously stated, PT can be implemented online given the numerical annotations. Moreover, it seems like "HIP-RL: Hallucinated Inputs for Preference-based Reinforcement Learning in Continuous Domains" can train a policy online.

---

### Official Review · Reviewer_fBpw · 2024-11-03

**Soundness:** 2
**Presentation:** 3
**Contribution:** 2
**Rating:** 3
**Confidence:** 4

**Summary:**

This paper presents an approach to use preference learning to solve a credit assignment. The main idea is to define 3 preference classes ([bad, neutral, good]) to evaluate the quality of an RL trajectory and, therefore, assign credit to episodic trajectories.

Through a series of experiments, the authors aim to demonstrate that their approach outperforms existing methods, including RRD, IRCR and gamma return.

**Strengths:**

The proposed trajectory preference-based credit assignment method is a novel and interesting approach. By defining three preferences ("good", "neutral", "bad") for each trajectory based on episodic rewards and the entire trajectory space, it provides a more comprehensive way to evaluate trajectories compared to existing methods.

**Weaknesses:**

1. Unclear motivation for using preference learning in credit assignments.
To me the motivation of using preference learning is unclear. Originally, preference-based RL is used when the ground truth of the reward signal is inaccessible. Therefore, we need to come up with an approach (the BT Preference model in such a case) to evaluate the reward without knowing its ground truth values. However, I do not see the intuitive benefit of using it in episodic RL.  I can give a very easy opposite example: if you can learn the classification between {bad, neutral, good} through the collected trajectories, why can't you do Value iteration with causal transformers? To me, learning the V(s) is essentially the same as learning the preference. I encourage the authors to justify their motivation clearly.

2. Unclear presentation
In the result section, such as Figure 2 and 3, the authors frequently use the term 'our method'. According to my understanding, the authors are proposing a new credit assignment method, not a new RL policy/value learning method. It is unclear to me what 'our method' means. Does it refer to a policy or a credit assignment method?

3. Unclear 'credit assignment' to performance boost
I find it difficult to believe the performance boost comes from the preference learning framework rather than the causal transformer structure. The causal transformer is well known as an excellent world model learner compared to MLP and RNN structures. It is reasonable to doubt that the causal transformer is so powerful to model the environment that preference learning does not really matter. I hope the authors can introduce ablation studies to justify the contribution of preference learning.

4. Insufficient experiments
The paper would be more convincing if the authors could demonstrate their method on at least 3 envs on Mojuco.

5. Questinable experiment setup and result for the sepsis task
The sepsis task is indeed a valuable dataset to study credit assignments, and I appreciate the authors mentioning it. The authors use a 'SOFA reward', which involves lactate and SOFA score. However, there are actually more alternative reward designs to test your method. For example, the original creators of the sepsis task used mortality as a reward [1]. [2] compared mortality, SOFA, and NEWS2 scores as rewards and found that the choice of reward can bias off-policy evaluation, making the doubly robust method 'doubly biased'. First, I encourage the authors to try other reward options, especially the mortality reward (outcome-only) and NEWS2 reward (-1~1 normalized) as 2 distinctive comparisons to study noisy reward. Secondly, since the doubly robust method is questioned in [2], the authors need to justify why WDR is used following an old paper and correct it if necessary.

[1] Komorowski, Matthieu, et al. "The artificial intelligence clinician learns optimal treatment strategies for sepsis in intensive care." Nature medicine 24.11 (2018): 1716-1720.
[2] Luo, Zhiyao, et al. "Position: reinforcement learning in dynamic treatment regimes needs critical reexamination." (ICML 2024).

**Questions:**

1. I want to challenge the definition of {bad, neutral, good} in terms of preference. How to make sure the Transitivity of Preferences holds? Does it really hold in a RL trajectory setting? Why can't one predict per-step reward, and do regression to its trajectory sum as the objective function?

2. Is it possible that a trajectory with good outcomes is assigned 'good' to all its steps, which loses the meaning of credit assignment? On the contrary, is it possible to find your model predicting 'bad' to most of the steps, but finally, leads to a good outcome, and vice versa? What does your classification mean then?

---

### Official Review · Reviewer_jj7M · 2024-11-06

**Soundness:** 1
**Presentation:** 3
**Contribution:** 2
**Rating:** 3
**Confidence:** 4

**Summary:**

This paper proposed a method to use preference-based RL method to learning a proxy reward for credit assignment. It first process the episodic return to get an tri-value ``global preference label''. Then it trains a causal transformer to predict this global preference label using state action sequences. After that, it use the output probability of transformer model on the correct global preference label as a reward factor (multiplied by the original episodic reward), and run standard RL methods with this reward.

**Strengths:**

1. This paper proposed some novel extension and applications for preference-based RL. It extends conventional PbRL by introducing global preferences to handle delayed and noisy rewards more effectively. This paper applies PbRL to the credit assignment problem.
2. The proposed method has potential real-world benefits to medical treatment (sepsis management) problems.

**Weaknesses:**

1. The second experimental domain, sepsis management in MiMIC III dataset, cannot provide significant support evidence to the benefit of proposed method. The problem is that the evaluation metric itself here is very noisy and unstable. Since online test is impossible, this paper use the weighted doubly robust methods, which can have a high variance. The paper lacks many analysis of the OPE estimator's, e.g.
 - confidence intervals from bootstrapping or concentration inequalities
 - the effective samples size of the importance weights in WDR
 - results and their consistency from multiple OPE estimators.
2. This proposed method lacks enough evaluation in more than one simulated environment. There are many simulated environment to test RL algorithm's performance with delayed and noisy rewards, e.g. different control problems in DeepMind control suite. Even within the Mujoco environment, there are more control tasks.
3. The proposed method needs to be compared with more reward shaping and intrinsic reward baselines, e.g., "On learning intrinsic rewards for policy gradient methods Z Zheng, J Oh, S Singh"

**Questions:**

1. It seems not clear to me why the proposed method is called preference based RL. The so-called global preference is actually discretization and normalization of the return. There is no preference or comparison between two or more actions/trajectories.
2. What is the difference between py  and P (py |st, at, τ1:t−1; φ) in Line 243? If py is defined as a probability, then P (py |st, at, τ1:t−1; φ) is a probability over probability which is hard to parse.

---

### Note · Authors · 2024-11-25

I have read and agree with the venue's withdrawal policy on behalf of myself and my co-authors.